## RESEARCH ARTICLE

# Tissue-specific transfer learning improves functional variant and therapeutic target discoveries in breast and prostate cancer

Qing Li[1,2]*, Dinghao Wang[3], Zilong Zhang[2], Deshan Perera[2], Zhishan Chen[1], Wanqing Wen[1], M. Ethan MacDonald[4,5,6,7,8], Weijia Cai[9,10,11], Jun Yan[12,13], Xiao-Ou Shu[1], Wei Zheng[1], Xingyi Guo[1]*, Quan Long[2,3,5,6,13]*

**1** Division of Epidemiology, Department of Medicine, Vanderbilt Epidemiology Center, Vanderbilt-Ingram Cancer Center, Vanderbilt University Medical Center, Nashville, Tennessee, United States of America, **2** Department of Biochemistry & Molecular Biology, University of Calgary, Calgary, Canada, **3** Department of Mathematics and Statistics, University of Calgary, Calgary, Canada, **4** Department of Electrical & Software Engineering, University of Calgary, Calgary, Canada, **5** Hotchkiss Brain Institute, University of Calgary, Calgary, Canada, **6** Alberta Children's Hospital Research Institute, University of Calgary, Calgary, Canada, **7** Department of Biomedical Engineering, University of Calgary, Calgary, Canada, **8** Department of Radiology, University of Calgary, Calgary, Canada, **9** Department of Medical Microbiology, Immunology and Cell Biology, Southern Illinois University School of Medicine, Springfield, Illinois, United States of America, **10** Simmons Cancer Institute, Southern Illinois University School of Medicine, Springfield, Illinois, United States of America, **11** Department of Surgery, Southern Illinois University School of Medicine, Springfield, Illinois, United States of America, **12** Physiology and Pharmacology, University of Calgary, Calgary, Alberta, Canada, **13** Department of Medical Genetics, University of Calgary, Calgary, Canada

* qing.li@vumc.org (QL); xingyi.guo@vumc.org (XG); quan.long@ucalgary.ca (QL)

## Abstract

DNA foundation models trained on large-scale genomic and epigenetic datasets have shown promise for regulatory variant interpretation, yet their application to tissue-specific contexts remain limited. Here, we present a transfer learning (TL) framework to adapt Enformer, a deep neural network trained on 5,313 multi-omics tracks, to breast and prostate cancer using 275 and 357 tissue-specific transcription factor (TF) ChIP–seq tracks, respectively. We computed tissue-specific cis-regulatory activity (tCRA) scores for millions of single-nucleotide variants (SNVs) in genome-wide association study (GWAS) datasets and prioritized high-impact SNV subsets (1M, 1.5M, and 2M). These TL-prioritized variants demonstrated consistently greater enrichment in tissue-specific enhancers, cancer GWAS risk variants, and ClinVar pathogenic variants compared to the original Enformer model. Transcriptome-wide association studies (TWAS) using TL-based SNVs identified more cancer-relevant genes, many of which exhibited functional essentiality (DepMap), therapeutic tractability (drug databases), and disease relevance (DisGeNET). Notably, TL models outperformed the base model in identifying genes enriched for drug targets and clinically relevant disease associations. Our results show that TL-derived tCRA scores enhance regulatory variant prioritization and improve susceptibility gene discovery in a tissue-specific manner. Our study provides a generalizable framework for tailoring

**Data availability statement:** Code availability The transfer learning and calculation code for tCRAs are available from Github website: https://github.com/theLongLab/Transfer-Learning. Data availability GWAS summary statistic data for breast cancer were downloaded from the BCAC website (https://www.ccge.medschl.cam.ac.uk/breast-cancer-association-consortium-bcac/data-data-access/summary-results). GWAS summary statistic data for prostate cancer were downloaded from PRACTICAL(https://practical.icr.ac.uk/?page_id=8164). Variants scores from Enformer were downloaded from: https://github.com/deepmind/deepmind-research/tree/master/enformer Cistrome TF ChIP-seq bed peaks files were downloaded from: http://cistrome.org/. Roadmap project 15 chromatin states were downloaded from https://egg2.wustl.edu/roadmap/web_portal/chr_state_learning.html#core_15state. Bed files for nine cell lines with 15 chromatin states from Roadmap project were downloaded from: https://egg2.wustl.edu/roadmap/data/byFileType/chromhmmSegmentations/ChmmModels/coreMarks/jointModel/final/. Bed files for prostate cell lines with 15 chromatin states: https://ngdc.cncb.ac.cn/omix/release/OMIX237. ClinVar variants annotation summary are downloaded from: https://ftp.ncbi.nlm.nih.gov/pub/clinvar/tab_delimited/variant_summary.txt.gz. Five diseases genes were downloaded from DisGeNET: https://www.disgenet.org/search with the following concept Unique Identifier (CUI): C0678222 (Breast Carcinoma), C3539878 (Triple Negative Breast Neoplasms), C0006142 (Malignant neoplasm of breast), C4722518 (Triple-Negative Breast Carcinoma) are used for breast cancer and C0600139 (Prostate carcinoma) and C0376358 (Malignant neoplasm of prostate) for prostate cancer. Gene essentiality scores were downloaded from DepMap: https://depmap.org/portal/. Drugs and their targets are retrieved from DrugBank (https://go.drugbank.com/), ChEMBL (https://www.ebi.ac.uk/chembl/) and therapeutic target databases (https://idrblab.net/ttd/). Enformer model weights: https://www.kaggle.com/models/deepmind/enformer.

**Funding:** This research was supported by the grant from US National Institutes of Health grant R37 CA227130 and R01 CA269589 to X.G. and a New Frontiers in Research Fund (NFRFE-2023-00291) and NSERC Discovery

foundation models to disease-relevant contexts, with implications for variant interpretation, therapeutic target discovery, and precision medicine.

## Author summary

Understanding how genetic changes contribute to cancer remains a central challenge in human genetics. While powerful deep learning models like Enformer can predict how DNA variants might affect gene regulation, they are often trained on very broad data and may not capture the tissue-specific mechanisms relevant to specific cancers. In this study, we developed a transfer learning (TL) approach to adapt Enformer for breast and prostate cancer by retraining it on datasets specific to each cancer type. This allowed us to compute regulatory scores for millions of genetic variants and identify those most likely to affect cancer risk. We found that our TL-enhanced models perform better at highlighting genetic variants located in tissue-specific regulatory regions. Using these high-priority variants, we linked genes to cancer risk through transcriptome-wide association studies (TWAS) and showed that many of the identified genes are important for cancer cell growth and are potential drug targets. Our findings demonstrate how adapting existing models to more disease-relevant data can significantly improve our ability to uncover genes and variants involved in cancer. This work provides a new tool for researchers aiming to understand genetic risk and discover future therapies.

## Introduction

Machine learning (ML), particularly deep learning, has transformed biological data analysis by enabling the integration of multi-scale omics data and the modeling of complex regulatory systems [1–6]. However, developing accurate ML models typically requires extensive training on large, high-quality datasets. In many specialized domains, such as tissue-specific or disease-focused analyses, such datasets are often unavailable or underpowered, limiting the applicability and relevance of general-purpose models. Moreover, Models trained on broadly heterogeneous datasets may capture signals not relevant to a particular tissue or disease, thereby underperforming in specialized applications. Transfer learning (TL) offers a powerful strategy to address this limitation [7–13]. TL generally involves adapting a general-purpose base model using a smaller, task-specific dataset through partial re-training of model parameters [14,15]. This approach allows researchers to redirect an existing model toward a specific biological question, leveraging prior training while improving relevance.

Genome-wide association studies (GWAS) have discovered thousands of disease loci [16–18], but causal variants and relevant tissues remain unclear for most. Towards this line, transcriptome-wide association studies (TWAS) improve GWAS by integrating expression quantitative trait loci (eQTLs), enabling tissue-specific gene

Grant (RGPIN-2024-04679) to Q.L. (Q. Long), D.P. was supported by an Alberta Innovates and an Eyes High scholarship. D.W. was supported by an Alberta Innovates scholarship. W.C. was supported by start-up funding from Southern Illinois University School of Medicine. The funders had no role in study design, data collection and analysis, decision to publish, or preparation of the manuscript.

**Competing interests:** The authors have declared that no competing interests exist.

discovery [19–21]. Recent studies have shown that the power of association mapping analysis can be significantly improved by integrating informative priors learned from existing transcriptomic and epigenetic data [22]. In the same vein, we have shown that TWAS can be substantially improved if prior knowledge from specific biological mechanisms (such as transcription factors, TFs) is integrated [23–25]. However, these efforts are segmented based on different prior knowledge, and there is currently no standardized framework for integrating diverse omics-based functional annotations in a tissue-specific manner to investigate cancer risk.

Enformer represents a significant advance in variant effect prediction by learning from over 5,000 functional genomic tracks across diverse human tissues and cell types [2]. By targeting the prediction of genetic variant effects on regulatory activities for each track of epigenetic profile, Enformer essentially provides integrated functional weights for any potential variants, which can be utilized by downstream analyses [26,27]. However, despite its comprehensive training data and highly sophisticated architecture, Enformer as a general-purpose model contains massive and heterogeneous information. As such, Enformer by itself may not be optimal for specific tasks in a target tissue.

Here, we present a transfer learning framework to redirect Enformer, a state-of-the-art deep learning model for regulatory variant effect prediction, toward breast and prostate cancer applications using tissue-specific transcription factor (TF) ChIP-seq data. We hypothesize that tailoring the model's regulatory priors to cancer-relevant TF binding patterns will improve the prioritization of functional variants and enhance downstream gene discovery. We chose breast and prostate cancer as two examples as our previous works show that alternation of tissue-specific TF-occupancy by cis-regulatory elements, is important to many genes and cancers [23–25,28,29]. In the following sections, we build two TL models, TL-breast and TL-prostate, using 275 and 357 TF ChIP-seq tracks, respectively. We derived tissue-based cis-regulatory activity (**tCRA**) scores for millions of variants from TL-breast, TL-prostate and their counterpart Enformer models, then defined high-impact SNV subsets and evaluated their functional relevance through enrichment analyses of enhancers, known GWAS risk variants, and pathogenic germline variants. Finally, we performed TWAS to identify candidate susceptibility genes and assessed their biological essentiality (via DepMap), therapeutic potential (via drug databases), and disease relevance (via DisGeNET).

## Results

### TL framework generated accurate statistical predictions

A general framework of Transfer Learning (TL) re-trains a subset of parameters in a pre-trained model based on a small number of dedicated data (Fig 1A). Along this line, our TL models are based on a pre-trained general-purpose model, Enformer (Fig 1B). We re-tasked Enformer to specificities of human transcription factor (TF) ChIP-seq data of tissue relevant tracks (Fig 1C) by selectively updating variety number of parameters. For breast cancer, we obtained tracks for 18 TFs (AR, CEBPB, E2F1, ESR1, FOSL2, FOXA1, FOXM1, GATA3, GREB1, HDAC2, JUND, NR2F2, PML,

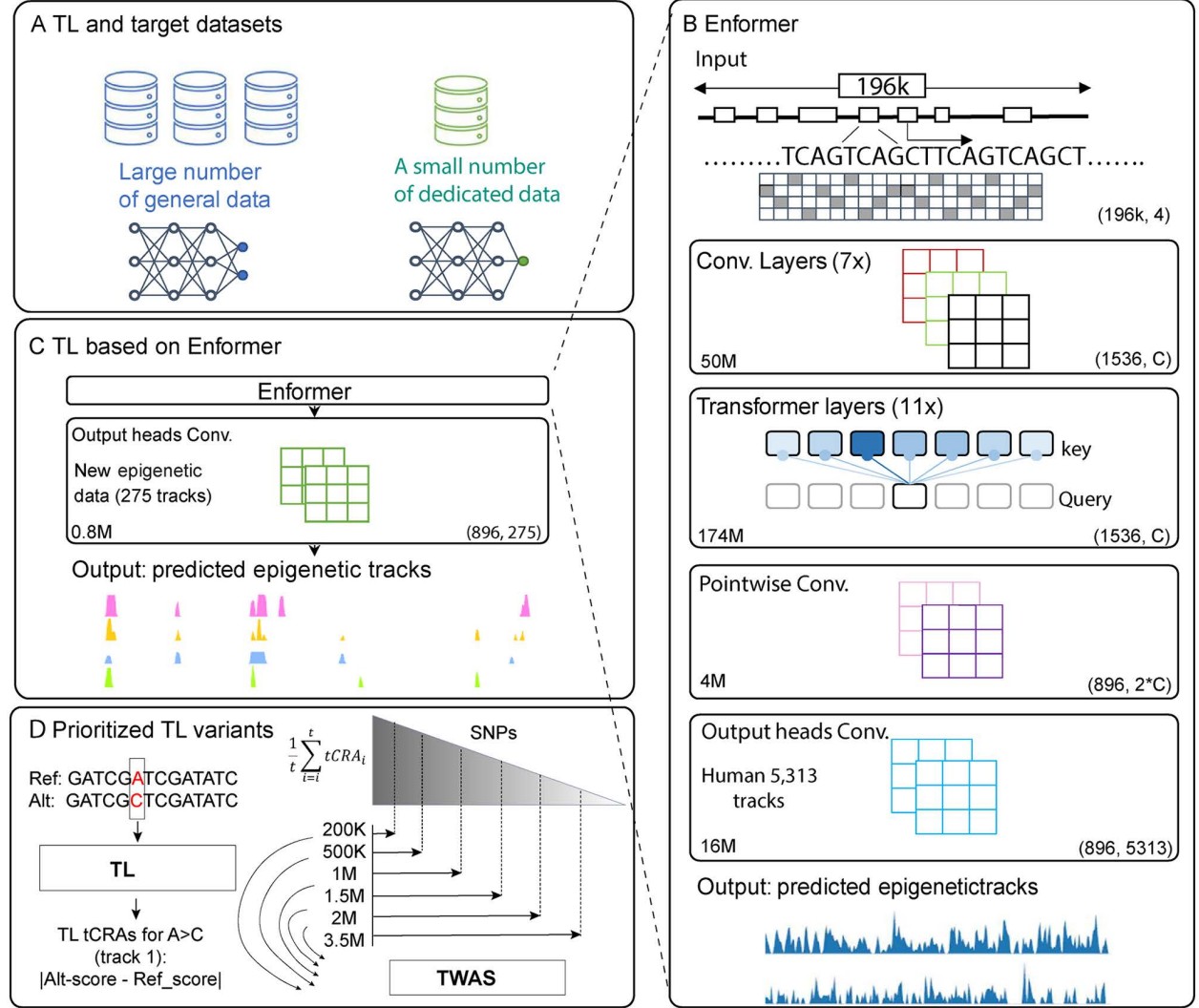

**Fig 1. The overview of the transfer learning workflow and construction of tCRAs. A)** A schematic of general TL: a small number of dedicated data may re-train a selected subset of nodes to redirect an existing model. **B)** Enformer model was trained on input DNA sequences (196kb) to predict multiple human epigenetic tracks (5,313) at 128-bp resolution. The network is composed of four blocks (indicated by each rectangle) from convolutional layers, transformer layers, pointwise convolution, and output heads convolution. The output shapes of these blocks are given by tuples on the bottom-right corner, in which C (1,536) indicates the number of channels in convolutional neural networks. The number of trainable parameters for each block is listed on the bottom-left corner. **C)** Our TL used majority of existing Enformer architecture together with its trained parameters by retaining its input and first three blocks. The only layer undergoes re-training is the output heads convolution layers, tailed to target-tissue epigenetic datasets (i.e., 275 tracks of TFs ChIP-seq for breast cancer). **D)** Left panel: An illustration of tCRAs for a specific genetic variant estimated by calculating the differences of predicted regulatory activity value between reference allele (A) and alternative allele (C) of a variant. Right panel: based on TL outcomes, we generated an activity score for each genetic variant for each track, which can be utilized in downstream analyses including association study.

SRF, TCF12, TCF7L2, TLE3, ZNF217), yielding a total of 275 tracks (S1 Table). For prostate cancer, we selected 17 TFs (AR, ARID1A, ASH2L, CREB1, ERG, FOXA1, GATA2, HOXB13, MED1, NANOG, NKX3–1, NR3C1, PIAS1, POLR2A, RELA, SUMO2, TLE3), totaling 357 tracks (S2 Table). Among all tracks, epithelium cell lines present the majority of them, with 172 (63%) from breast and 331 (92%) from prostate. The Enformer model redirected to TF ChIP-seq data from breast is denoted as TL-breast (S1 Fig). Similarly, the model redirected to TF ChIP-seq data from prostate is denoted as

TL-prostate (S1 Fig). Among these ChIP-seq tracks, pairwise correlation analyses (Materials and Methods) showed presence of substantial biological heterogeneity (S3–S4 Tables), with tracks derived from the same cell line exhibiting significantly higher concordance than tracks originating from different cell lines (S2 Fig).

The transfer learning (TL) framework generated statistically reliable predictions, with accuracy assessed by the correlation between predicted and observed TF ChIP-seq tracks. We calculated the Pearson correlation coefficients for 18 breast TFs (Fig 2A) and 17 prostate TFs (Fig 2B) on held-out test datasets. On average, correlations were around 0.42 for breast TFs and around 0.47 for prostate TFs, supporting the ability of the TL model to capture essential peak-level information and demonstrating successful adaptation of Enformer to tissue-specific contexts. To further evaluate consistency between the TL models and Enformer on matched inputs, we compared their predictions across 1,937 input DNA sequences from test datasets per transcription factor (Materials and Methods). POLR2A was the only TF shared between the two models, with 42 tracks in the TL-prostate model (S2 Table) and four corresponding tracks in Enformer (S5 Table). Using this TF, we observed a notable concordance between model outputs, with an average Pearson correlation of 0.55 (S6 Table), indicating that the TL-prostate model preserves much of Enformer's regulatory signal for POLR2A while incorporating tissue-specific refinements introduced during transfer learning. In addition, we observed considerable variability in prediction performance across TF ChIP-seq tracks, particularly for ESR1 ($r = 0.10$-$0.75$) and AR ($r = 0.08$-$0.60$), suggesting heterogeneity across cell lines or experimental platforms. We stratified tracks into three correlation-based groups (low, moderate, and high) and evaluated their associations with standard ChIP-seq quality metrics (Materials and Methods). For breast, 275 tracks were distributed into 71 (low), 167 (moderate), and 37 (high). For prostate, 357 tracks were distributed into 40 (low), 250 (moderate), and 67 (high). In both datasets, general sequencing metrics (FastQC score, uniquely mapped read ratio, and PCR bottleneck coefficient) did not differ substantially across performance groups (S3A–S3C and S4A–S4C Figs).

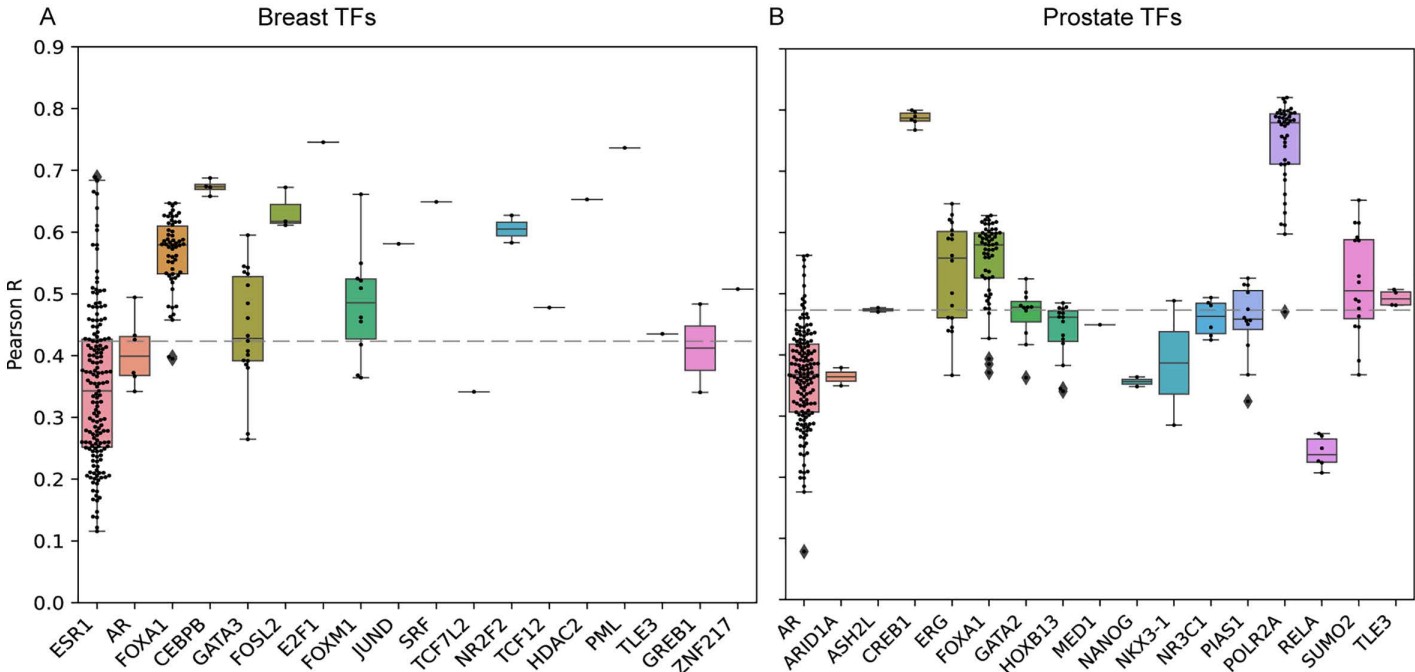

**Fig 2. Performance of TL in statistical predictions. A)** Box plot to show the average Pearson correlation coefficients (by the number of tracks) between predicted outputs and true outputs for each TF. The grey dash line is the average Pearson correlation across TFs. **B)** The same as A, but for prostate TFs.

In contrast, peak-enrichment metrics (PeaksFoldChangeAbove10, FRiP, and union DHS ratio) were significantly lower in low-correlation tracks compared to moderate and high groups in breast (S3D–S3F Fig), with similar trends observed in prostate (S4D–S4E Fig).

## SNVs prioritized by TL-derived tCRAs are enriched in functional annotations

A fundamental hypothesis behind developing TL models is that TL models have enhanced tissue- and disease-specific performance compared to the general-purpose Enformer. To evaluate this, we prioritized three sets of genetic variants (1M, 1.5M, 2M) based on tCRA scores derived from the TL-breast and TL-prostate models and compared their functional relevance to SNVs prioritized by Enformer(Materials and Methods). Our results consistently support this assumption from three perspectives. First, our results showed SNVs prioritized by TL-breast exhibiting significantly greater enrichment in breast cell line–specific enhancer regions compared to those prioritized by Enformer: 13.3% vs. 7.5% for the top 1M SNVs, 11.7% vs. 7.1% for the top 1.5M, and 10.6% vs. 7.6% for the top 2M (Fig 3A and S7 Table). Similarly, TL-prostate prioritized SNVs were more enriched in prostate cell line–specific enhancers than those from Enformer: 15.8% vs. 10.2% for 1M, 13.6% vs. 10% for 1.5M, and 12.0% vs. 9.5% for 2M (Fig 3B and S7 Table). Next, we stratified the enhancer-overlapping SNVs by cell type. TL-breast SNVs exhibited a higher proportion of overlap in breast-related cells than in non-breast cells. Although a similar pattern is observed in the Enformer prioritized SNV subsets, the difference is less pronounced than in the TL case (Fig 3C and S7 Table). A similar pattern was observed for prostate cancer: TL-prioritized SNVs showed greater differentiation between prostate and non-prostate cell types compared to Enformer (Fig 3D and S7 Table). Moreover, we consistently observed the stronger enrichment of GWAS risk variants in TL prioritized SNV sets compared with those prioritized by Enformer (Fig 3E and S7 Table). A similar trend was observed in ClinVar annotations: the number of pathogenic SNVs identified was 14 versus 6 for TL and Enformer at the 1M threshold, 18 versus 11 at the 1.5M threshold, and 23 versus 18 at the 2M threshold (Fig 3F and S7 Table). Together, these results demonstrate that our TL framework effectively adapts the general-purpose Enformer model to tissue-specific TF binding profiles, thereby improving the identification of regulatory and disease-relevant variants.

## TL-prioritized variants improve genotype-disease association mapping in TWAS

As a model of integrating gene expression data in GWAS, TWAS analyzes the correlation between disease phenotype and an aggregation of cis-genetic variants selected by gene expressions [20,30]. A conventional TWAS protocol usually uses a regularized multiple regression (e.g., elastic net) to select SNVs and their weights in the form of training a gene expression prediction model (in a reference dataset such as GTEx [31]; and then apply this model to "impute" the expression in the GWAS dataset for assessing the association (Materials and Methods). We performed TWAS using the full set of 9.9 million SNVs as a baseline. Our results showed 62 and 70 genome-wide significant genes from genetically well predicted gene models ($R^2 > 0.01$) breast and prostate cancer, respectively, after Bonferroni correction (S8 Table).

To evaluate the association of tCRA-prioritized SNVs subsets (1M, 1.5M and 2M) and cancer risks, we conducted TWAS analyses using SNVs subsets derived from TL models (Materials and Methods). Overall, TL-based tCRA subsets enabled the identification of more predictable genes ($R^2 > 0.01$) than Enformer-based subsets (S9 Table). At Bonferroni-corrected $P < 0.05$, TL-prostate models identified more genes than Enformer in 1.5M and 2M subsets, while TL-breast identified slightly fewer genes than Enformer (S10 Table). Among these genes, three genes (*RPS23*, *RPLP2* and *SUGP1*) from TL-breast three subsets play critical roles in cell proliferation (Fig 4A), as shown by their median CERES score less than −0.5 from DepMap. Enformer three subsets also captured *RPS23*, *RPLP2* but missed *SUGP1* (S11 Table). As for prostate, TL-prostate three subsets exhibiting five genes (*SF3B4, MAT2A, GEMIN4, WTAP, GMPPB*) play critical roles in cell proliferation (Fig 4B), while Enformer three subsets also reveal six critical genes, but missed *SF3B4* and instead reported *TCP1 and ADAR* (S12 Table). In addition, we also evaluated the therapeutic relevance of these genes by identifying druggable targets and found TL models consistently identified more druggable targets linked by approve or drugs in

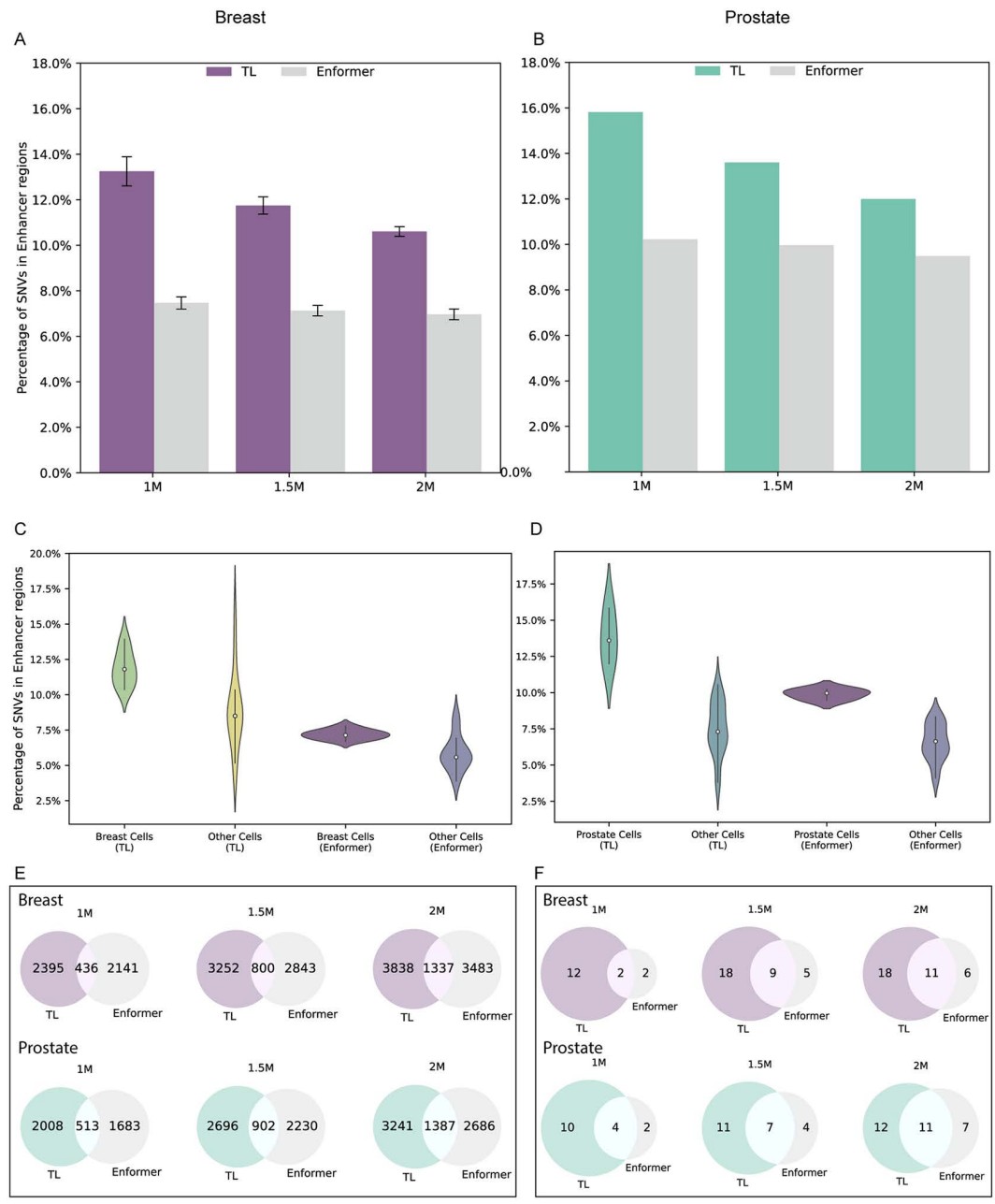

**Fig 3. Functional enrichment of six SNVs subsets. A)** The proportion of SNVs located within enhancer regions for the three TL-breast or Enformer-prioritized SNV subsets, using their corresponding tCRA scores. Intervals were calculated by three breast cell lines. **B)** The same as A but evaluating SNVs prioritized from TL-prostate and Enformer on the prostate cell line. **C)** Violin plots for proportion of SNVs located in enhancer regions of all TL-breast and Enformer prioritized tCRAs within breast-specific cell lines and other cell lines. **D)** The same as C but with prostate-specific cell and other cell lines. **E)** Venn plots for number of SNVs show GWAS significance ($P < 5 \times 10^{-8}$) from 1M, 1.5M and 2M sets of genetic variants from breast-TL and breast-Enformer model (top row), as well as prostate-TL and prostate-Enformer model (bottom row). **F)** The same as E but the SNVs annotated as pathogenic in ClinVar.

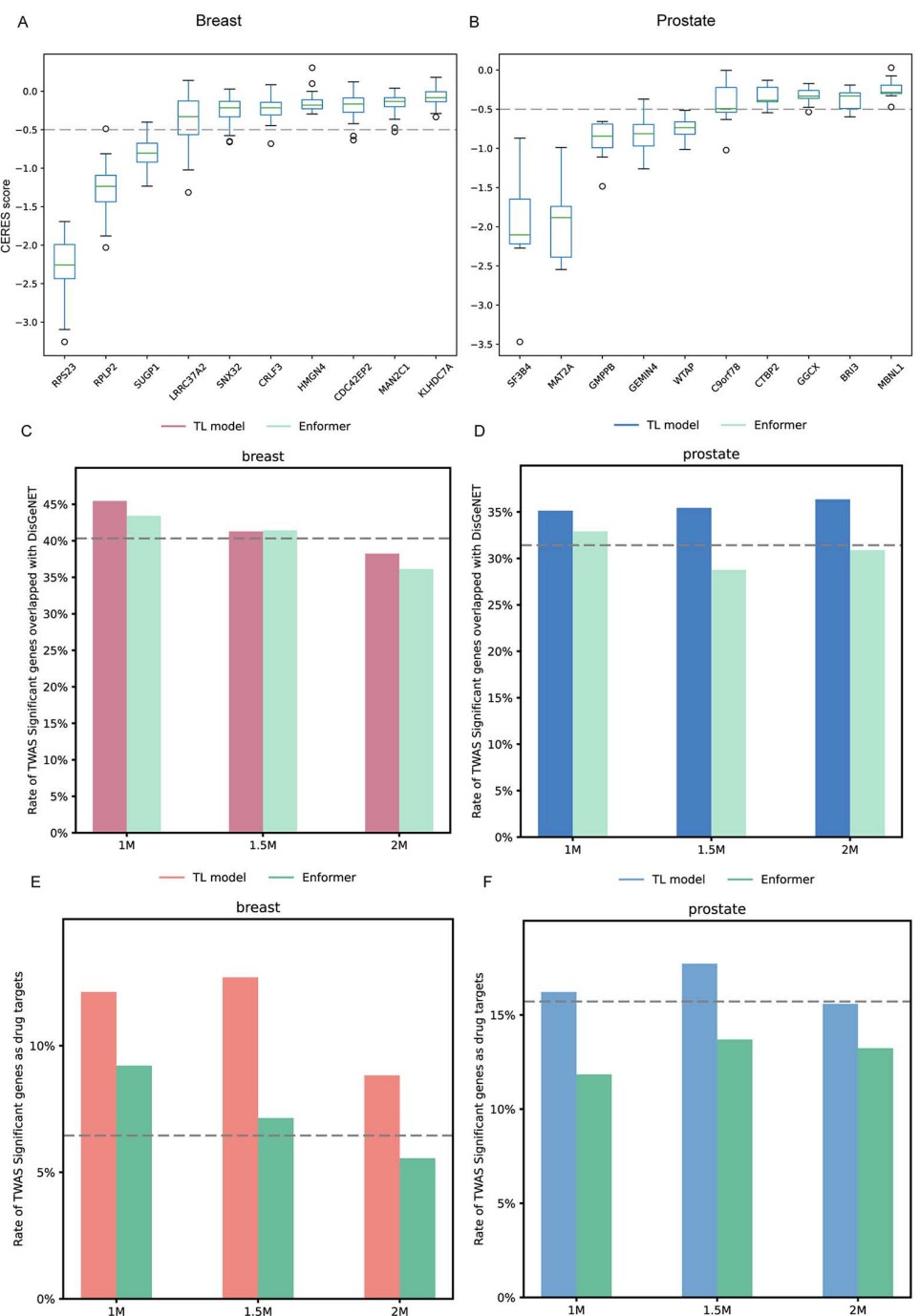

**Fig 4. Biological and therapeutic relevance for TWAS significant genes. A)** Boxplots depicting top 10 genes dependency (CERES) scores derived from breast cancer (48 cell lines) CRISPR-Cas9 experiments (DepMap 23Q4) for TL-breast model three SNVs subsets. The dashed line represents a suggested CERES value cutoff of −0.5, below which indicates a higher dependency of cell proliferation on the gene. **B)** The same as A, but from prostate cancer (9 cell lines) for genes identified by the TL-prostate model. **C)** Bars plot to show percentage of TWAS significant genes as drugs targets, utilizing TL-breast or Enformer-prioritized subsets of SNVs. **D)** The same as **C)** but for TWAS results using TL-prostate or Enformer-prioritized subsets of SNVs. **E)** DisGeNET enrichment rate (percentage of TWAS-significant genes found in DisGeNET) for TL-breast and Enformer models across the three SNV subsets. **F)** The same as **E)** but for prostate cancer. Grey dashed line refers to the baseline TWAS model result in **C-E**.

clinical trial phase II/III and showed higher enrichment of druggable genes than Enformer. For example, in the TL-breast 1M subset, eight druggable genes were identified versus seven in the Enformer subset (S10 Table). For the 1.5M and 2M subsets, TL-breast identified eight and six druggable genes, respectively, compared to five and four from Enformer. The enrichment rate of druggable genes in the TL-breast model was also higher than in Enformer (e.g., 12% vs. 9% in the 1M subset, Fig 4C). In prostate cancer, TL-prostate outperformed Enformer in both the number and proportion of druggable genes across all subsets. Specifically, TL-prostate identified 12, 14, and 12 druggable genes in the 1M, 1.5M, and 2M subsets, respectively, compared to 9, 10, and 9 in Enformer. The enrichment rate of druggable genes in the TL-breast model was also higher than in Enformer (e.g., 16% vs. 12% in the 1M subset, Fig 4D). Furthermore, we found 54 drugs, either approved or under clinical trials, targeting the eight druggable genes identified by the TL-breast 1M subset, and 242 drugs targeting the 12 genes identified in the TL-prostate 1M subset (S13 Table). Finally, we assessed the disease relevance of TWAS-identified genes using DisGeNET. Both TL-breast and TL-prostate consistently showed higher DisGeNET enrichment rates than Enformer across all subsets (Fig 4 E and 4F). Collectively, our results demonstrate that TL-derived tCRAs have incremental effect on the identification of biologically and clinically relevant disease susceptibility genes and offer more promising candidates for therapeutic targeting for breast and prostate cancer.

## Discussion

In this study, we applied transfer learning (TL) framework to redirect Enformer models to tissue-specific TL models and enhanced regulatory genetic variants and cancer suspicious genes discovery. Using the TL-derived tissue-based cis-regulatory activity (tCRA) scores, we prioritized millions of variants and defined high-impact SNV subsets for functional evaluation and gene discovery. Compared to the Enformer model variants, TL-prioritized variants demonstrated consistently higher enrichment in tissue-specific enhancer regions (e.g., 13.3% vs. 7.5% in breast at 1M SNVs), GWAS-identified cancer risk loci, and ClinVar pathogenic variants (14 vs. 4 in breast at 1M SNVs). TWAS using TL-prioritized variants identified more cancer-relevant genes, with several showing essentiality based on CRISPR-Cas9 dependency screens and therapeutic relevance in drug target (12% vs. 9% in breast at 1M SNVs) and disease association databases (45% vs. 43% in breast at 1M SNVs). Taken together, our results highlight the utility of transfer learning in tailoring general-purpose regulatory models to disease-specific contexts.

A key novelty of our framework lies in its application of transfer learning. Functional annotation using epigenetic data from non-target tissues can introduce signals unrelated to the disease context, creating a fundamental trade-off in multi-scale modeling: excessively heterogeneous data may dilute disease-relevant biology, whereas overly narrow models limit generalizability. Our transfer learning strategy addresses this challenge through a two-step design. First, a foundation model is trained on diverse multi-omic data to capture broad regulatory features. Second, this general regulatory model is redirected toward the appropriate biological context by fine-tuning with tissue- or disease-specific TF ChIP-seq profiles. Empirically, we also demonstrate the necessity of transfer learning. A simple convolutional neural network (CNN) trained from scratch on breast TF ChIP-seq data failed to learn long-range regulatory dependencies and performed substantially worse than the TL-breast model on held-out data (Pearson $r = 0.18$ vs. 0.39; S5 and S6 Figs). Functional enrichment was similarly weaker for the CNN across the top 1M, 1.5M, and 2M SNP subsets (CNN vs. TL-breast: 1M subset (7.5% vs. 13.3%), 1.5M subset (7.3% vs. 11.7%) and 2M subset (7.1% vs. 10.6%)), underscoring its limited ability to capture biologically meaningful regulatory signals. Moreover, transfer learning also confers substantial computational advantages: fine-tuning of TLs was completed within three days on a single NVIDIA A100 GPU, whereas training Enformer from scratch requires 64 TPU v3 cores for a similar duration. Thus, transfer learning offers a practical and efficient strategy for leveraging rich regulatory knowledge embedded in pretrained foundational models while adapting them to disease-relevant cellular contexts.

Another key strength of our framework is that it allows us to prioritize variants with functional relevance synergically. Unlike traditional TWAS approaches that select SNVs solely based on genotype–expression correlations, our method restricts predictors to SNVs enriched for regulatory activity, thereby enhancing statistical power and improving

interpretability by anchoring TWAS associations to experimentally informed regulatory elements. Nevertheless, the use of TL-derived SNVs mainly dependent on the completeness and accuracy of the external model, which may introduce tissue- or assay-specific. Although our enhancer-enrichment analyses support the functional relevance of prioritized variants, complementary approaches will be needed to more fully characterize their regulatory architecture and ensure that model-derived SNVs capture the full spectrum of biologically meaningful mechanisms.

The applications of our fine-tuned models are not limited to TWAS demonstrated in this work. All the applications that require interpretation of variants may benefit from this fine-tuned model. For instance, fine-mapping is a promising field where the integration of prior knowledge-based scores can play a crucial role [22,26]. Additionally, traditional Mendelian randomization Materials and Methods, which rely on clumping and thresholding to select instrumental variants, can also incorporate tCRAs scores to enhance the detection of causal directions. Furthermore, these scores can be utilized to train more accurate polygenic risk scores for clinical use, improving the predictive power and utility of these scores [32,33] in personalized medicine. While some TL models SNVs subsets identified not as many genes as Enformer, they consistently showed higher enrichment in therapeutic targets and disease evidence genes from DisGeNET, suggesting that TL-derived TWAS results are more clinically relevant. Many of drugs, targeting cancer risk genes, are currently approved for non-cancer indications, they hold potential for repurposing in cancer therapy due to their targeting of genes implicated in breast or prostate cancer risk. An interesting future work could involve integrating this framework with theoretical research on the transferability [34] of existing models to quantify the feasibility of such transfer learning and automate some of the parameter tuning steps.

Nevertheless, we acknowledge several limitations of this study. First, our framework utilized only TF ChIP-seq tracks for fine-tuning, while other tissue-specific epigenetic modalities, such as ATAC-seq or histone modification profiles, could provide complementary insights into chromatin accessibility and enhancer activity, potentially enhancing variant interpretation. Second, the TF ChIP-seq tracks used for transfer learning include mixtures of biological and technical replicates, contributing to heterogeneity across tracks. Tracks from the same cell line generally showed more consistent behavior, whereas tracks from different cell lines displayed markedly greater divergence. Future efforts to disentangle replicate structure may further improve the characterization of TF-specific regulatory signals for model training. Third, the substantial variability in Pearson correlation observed for ESR1 and AR likely reflects biological and technical heterogeneity rather than model limitations. Notably, low-correlation tracks displayed reduced peak-enrichment metrics (PeaksFoldChangeAbove10 and FRiP) compared to moderate and high groups, underscoring the importance of peak-strength measures in track selection. These findings suggest that transfer learning, given its limited fine-tuning steps, may require more stringent quality thresholds and careful curation of ChIP-seq tracks than conventional training approaches. Moreover, although our approach significantly enhances tissue relevance by redirecting Enformer, the state-of-the-art model at the time of our analysis, using disease-specific TF binding data, we did not benchmark its performance against a broader set of more recently developed regulatory models. Future work could expand this comparison by including emerging architectures such as Borzoi [35], to further contextualize the utility and limitations of our transfer learning framework. Lastly, while the transfer learning framework enables efficient model adaptation, it still requires manual curation and empirical evaluation to optimize fine-tuning performance for specific applications. Therefore, continued development of automated strategies for model selection, parameter tuning, and domain adaptation will be critical for fully realizing the potential of transfer learning in regulatory genomics.

## Materials and methods

### Selection of TF ChIP-seq data

Our previous studies identified multiple risk TFs with central roles in regulating gene expression and significantly associated with breast and prostate cancer [23,25]. Building on these findings, we systematically queried the Cistrome database

[36] and curated high-quality ChIP-seq tracks corresponding to risk TFs. We retained only tracks that passed quality control thresholds suggested by Cistrome (FastQC score > 25; uniquely mapped read ratio > 0.6; PCR bottleneck coefficient > 0.8; PeaksFoldChangeAbove10 ≥ 500; fraction of reads in peaks (FRiP) > 0.01; union DNase I hypersensitive (DHS) ratio > 0.7). To ensure robustness, we included multiple ChIP-seq tracks per TF, enabling the deep learning model to capture consistent binding patterns across experiments while tolerating variability from differences in cell lines, experimental conditions, or batch effects. Each TFcell line combination was treated as a distinct biological context and retained as an independent training track, and independent experiments profiling the same TF in the same cell line were also retained individually. Technical replicates could not be explicitly identified from Cistrome metadata.

## Pairwise correlation of ChIP–seq tracks

To assess biological concordance among ChIP–seq tracks corresponding to the same TF, we collected all available tracks for each TF and quantified their similarity using the original ChIP–seq signal intensities. Pairwise correlations were computed between all track pairs based on 1,937 sequences targets from the held-out test dataset. For each track pair, Pearson correlation coefficients were calculated on a per-sequence basis and subsequently averaged across 1,937 sequences to generate an overall measure of track-level concordance for each TF.

## Data preparation

We followed the Enformer framework to construct training, validation, and test datasets. The dataset includes 34021 training, 2213 validation, and 1937 test samples. Each sample consists of an input DNA sequence (196,608), $k$ tracks output of corresponding ChIP-seq targets with 896 values for each track. To prepare input DNA sequences, following Enforemer tutorial, we use 131072-bp regions predefined by Basenji2 and extend them to 196608 bp regions, extract DNA sequne from the reference genome (hg38.ml.fa), using one-hot coded to generate 196608 x 4 matrix. The output targets are tissue-specific TF ChIP-seq tracks derived from ChIP-seq data mapped to hg38 and stored in bigWig format. These tracks were aligned to the same genomic positions as the DNA inputs, cropped by 40960 bp from each end to avoid edge effects. To mitigate large-scale discrepancies across tracks, output targets are normalized and clipped to a maximum value of 32, scaled by 2 (for numerical stability), and averaged across 128 positions to yield predictions at 128-bp resolution, resulting in 896 values (196608–40960*2)/128, 128 bp per bin) per trac. These preprocessing parameters were selected to be consistent with the TF ChIP-seq processing strategy described in Enformer's S2 Table. Continuous TF ChIP-seq signals were used because they preserve binding-strength information, provide richer quantitative resolution, and is more informative for deep learning models than sparse binary peak labels. Therefore, in this study, one sample for TL models contains inputs are one-hot-encoded DNA sequences (A = [1,0,0,0], C = [0,1,0,0], G = [0,0,1,0], T = [0,0,0,1], N = [0,0,0,0]) and outputs are matrixes of size 896 × $k$ ($k$ = 275 for breast or 357 for prostate), representing predicted TF binding profiles across the genomic region.

## Transfer learning (TL) model architecture

A general TL model can be described as $M = (T, D, \Theta)$, where $T$ is the network topology, $D$ is the training dataset, and $\Theta$ is the set of parameters optimized using a loss function. The TL model $M^{TL} = (T', D', \Theta_0, \Theta')$ modifies a subset of nodes in $T$ leading to the revised network $T'$; and supply additional data ($D'$) to re-train a subset of the parameters $\Theta'$, retaining the rest parameters $\Theta_0 = \Theta - \Theta'$. The size of $\Theta'$ indicates how authentic the new model will be consistent to the original model. Enformer [2] is a profoundly advanced deep neural network with self-attention mechanism, comprising ~200 million parameters, trained on human and mouse genomic sequences and human and mouse epigenetic profiles. As for human tasks, it uses 196,608-bp input DNA segments from the hg38 reference genome and predicts over 5,313 genomic tracks from transcription factors (TF) binding, chromatin marks, and transcription. The architecture comprises four modules: ($T_1$) convolutional blocks with pooling, ($T_2$) transformer blocks, ($T_3$) a cropping layer with pointwise convolutions, and ($T_4$)

output heads for organism-specific predictions. Our TL models retain the core architecture of Enformer and replace the organism-specific heads ($T_4$) with a new output layer tailored to $k$ tissue-specific ChIP-seq tracks. We also added a dense layer, resulting in $k$ output channels, each corresponding to a distinct track. This is implemented using snt.Linear function, followed by a softplus activation function, as in Enformer. The TL-breast model and TL-prostate model are presented in S1 Fig. The number of trainable parameters in TL models ranges from 20M to 30M.

### Hyperparameter setting and TL model training

We initiated the TL-breast, TL-prostate model with Enformer well-trained weights and bias and only fine-tuned a few layers closer to the output. We incorporated many hyperparameters from Enformer training process (e.g., batch size equaling to 64 and *Adam* optimizer with a learning rate of 0.0001). Moreover, TL framework has an additional hyperparameter that is the number of layers to be fine-tuned. We meticulously explored the number of layers to be fine-tuned during the transfer learning process and we explored four fine-tuning configurations (S1 Fig): 1) updating parameters only in the final linear fully connected layer; 2) updating the final layer and the pointwise convolutional block; 3) updating the final layer, the pointwise convolutional block, and one transformer block; 4) updating the final layer, the pointwise convolutional block, and eleven transformer blocks. The model was trained to predict ChIP-seq signals using a Poisson negative log-likelihood loss function (Equation 1). We performed our training using Sonnet v2 and TensorFlow v2.4.1 on a single A100 GPU. We applied the early stop strategy, which stopped transfer learning when the decrease in loss on the validation dataset was less than 0.001 for five consecutive steps, thereby preserving the knowledge previously acquired from Enformer. We selected the model that minimized the loss function. Finally, we utilized the Pearson correlation between the predicted and actual values in the new task (i.e., the ChIP-Seq tracks of breast and prostate) as the indication of whether transfer learning is successful.

$$L_{Poisson}\left(T'\left(d;\ \Theta'\right),\ f(C)\right)=\ T'\left(d;\ \Theta'\right)-f(C)\cdot log\left(T'\left(d;\ \Theta'\right)\right)$$

(1)

Here, $d \in D'$ denotes an input DNA sequence fragment, and $T'(d;\ \Theta')$ is the predicted ChIP-seq signal produced by the TL-model $T'$ with trainable parameters $\Theta'$. The observed target $f(C)$ represents the preprocessed ChIP-seq signal corresponding to the same genomic region as $d$, obtained via a preprocessing function $f(\cdot)$ applied to the raw ChIP-seq data $C$, This loss function is used to fine-tune the target parameters $\Theta'$ of in the model $T'$.

### Performance evaluation of TL models

To evaluate the performance of TL models' training, we calculated Pearson correlation coefficient ($r$) between the predicted and actual track profile based on 1,937 DNA sequences from the held-out test datasets. For each DNA sequence, the output is a matrix of size $896 \times k$, with each representing the predicted value. For each of the $k$ tracks, we calculated the Pearson correlation coefficient $r$ by comparing two vectors of length $1,937 \times 896$: one containing the true ChIP-seq values and the other containing the corresponding predictions from the TL model.

### Evaluation of ChIP-seq tracks quality metrics and their prediction performance

Based ont the Pearson correlation coefficient performance, tracks were stratified into three groups: low ($r < 0.3$), moderate ($0.3 \le r \le 0.6$), and high ($r > 0.6$). To assess whether ChIP-seq quality metrics were associated with prediction performance, we compared the distributions of six Cistrome-reported quality metrics (FastQC score, uniquely mapped read ratio, PCR bottleneck coefficient, PeaksFoldChangeAbove10, fraction of reads in peaks (FRiP), and union DNase I hypersensitive (DHS) ratio) across the three groups. Pairwise differences (low vs. moderate, low vs. high, moderate vs. high) were evaluated using the Kolmogorov–Smirnov (KS) test. We considered a significant difference (KS $P < 1 \times 10^{-5}$) as evidence that the corresponding quality metric is associated with differences in prediction performance between groups.

## Performance comparison of TL and Enformer models

To evaluate the consistency between the TL models and the original Enformer model, we compared their predictions for overlapping TFs using identical input DNA sequences. For each tissue, we identified TFs present in both the TL model and Enformer. To reduce track-level heterogeneity, predictions were averaged across all tracks corresponding to the same TF within each model, yielding a vector of 896 output values per model for each input sequence. We calculated the Pearson correlations between the TL-model and Enformer prediction vectors across 1,973 DNA sequences from the held-out test dataset.

## Baseline convolutional neural network trained from scratch

To benchmark the necessity of pretrained regulatory knowledge from Enformer, we constructed a baseline convolutional neural network trained from scratch to predict TF ChIP–seq signals directly from DNA sequence. The CNN architecture consisted of four convolutional layers (128, 256, 512, and 512 channels; kernel size = 15; strides = 2, 4, 4, and 4), followed by a final convolutional projection layer. Input sequences of 196,608 bp were symmetrically cropped to 114,688 bp and progressively down sampled to yield an output tensor of 896 bins × $k$ tracks, where $k$ is the number of TF ChIP–seq datasets ($k$ = 275 for breast TFs). The CNN was trained de novo using the same optimization settings as the TL models (Adam optimizer, learning rate = $1 \times 10^{-4}$, batch size = 64, Poisson negative log-likelihood loss) on an NVIDIA Tesla V100 GPU. To ensure a fair comparison, training time was capped at three days, matching the training time of TL models. Model performance was evaluated using Pearson correlation between predicted and observed ChIP–seq tracks on held-out validation and test sets.

## Construction of Tissue-based Cis-Regulatory Activities (tCRAs)

A key application of Enformer is estimating the influence of genetic variants on target track profiles. TL models naturally inherit these properties and can predict impact of genetic variants on tissue-specific TF activities, which we termed as target **t**issue-based **C**is-**R**egulatory **A**ctivities (**tCRA**s), calculated through mathematical formula (Equation 2).

$$tCRA_i = \frac{1}{k} \sum_{j=1}^{k} \left| Pred_{i,j}^{alt} - Pred_{i,j}^{ref} \right|$$

(2)

where, $tCRA_i$ represents the regulatory activity of the $i$-th SNV, defined as the absolute mean difference between the predicted values for the alternative allele ($Pred_{i,j}^{alt}$) and reference allele ($Pred_{i,j}^{ref}$) across all tracks ($j$). These predictions are made using a 196,608-bp input DNA sequence centered on the SNV and produce an output vector of 896 values (128-bp bins) on $k$ tracks. We positioned SNV one base to the right of the DNA sequence center. We computed the predicted effect as the difference between the values at the center of the output, by averaging the 448th and 449th bins. The regulatory effect of $i$-th SNV on $j$-th track is defined as the absolute difference between $Pred_{i,j}^{ref}$ from $Pred_{i,j}^{alt}$. To compare TL models with the original Enformer, we downloaded precomputed variant effect predictions from the Enformer GitHub repository and applied the same formula to calculate tCRAs using Enformer's tissue-specific output tracks (S3 Table).

## SNVs prioritization using tCRAs

Since tCRAs indicate the extent to which single nucleotide variants (SNVs) affect TF ChIP-seq profiles, one intuitive application is to use them to prioritize SNVs that may have significant functional roles. With this in mind, we calculated tCRA scores for millions of SNVs from breast cancer GWAS summary statistics, obtained from the Breast Cancer Association Consortium (BCAC, N = 128,951; 122,977 cases and 105,974 controls) [37] and prostate cancer GWAS summary statistics data downloaded from the Prostate Cancer Association Group Investigate Cancer Associated Alterations in the

Genome (PRACTICAL, N = 140,306; 79,194 cases and 61,112 controls) [38]. SNVs exhibit Hardy-Weinberg equilibrium and minor allele frequency > 0.01 (9,898,466 for breast and 9,702,549 for prostate) are considered as passed quality control and their tCRAs values are estimated from TL models and Enformer. Next, we ordered SNVs by their mean tCRAs values across all tracks in TL models and prioritized top variants to form subsets, guided by the intuition that SNVs with higher regulatory discrepancy between reference and alternative allele play more critical roles than those with lower discrepancy. For the Enformer model, we applied the same strategy utilizing tCRAs calculated from risk TF ChIP-seq tracks from breast and prostate cancer cell lines. This resulted in three prioritized SNV subsets for each model: TL-breast, TL-prostate, and Enformer (breast and prostate), corresponding to 1M, 1.5M and 2M SNVs, respectively.

## Functional evaluation of SNV subsets

To comprehensively assess the functional relevance of SNVs prioritized by the TL models and Enformer, we evaluated three aspects: (1) enrichment of prioritized SNVs in tissue-specific enhancer regions; (2) differential enrichment between enhancer regions of tissue-relevant versus unrelated cell lines; (3) overlap with known breast and prostate cancer GWAS risk variants and pathogenic germline variants from ClinVar [39]. For enhancer enrichment, we used chromatin state annotations from four tissue-derived cell lines (three breast cell lines from the Roadmap Epigenomics Project and one prostate cell line from Roadmap Epigenomics Project [40] and one prostate cell line from [41] annotated across 15 chromatin states. To assess tissue specificity, we compared enrichment against six randomly selected, unrelated cell lines from the Roadmap Epigenomics Project (S14 Table). For GWAS overlap, we retrieved $P$ values from summary statistics and counted SNVs reaching genome-wide significance ($P < 5 \times 10^{-8}$) within each prioritized subset. In parallel, we downloaded variant annotations from ClinVar, retaining only those annotated "pathogenic" and "germline," and quantified their representation in the TL and Enformer subsets.

## Transcriptome-wide association study (TWAS)

To further demonstrate the utility of tCRA-based variant prioritization, we performed TWAS using gene expression and whole-genome sequencing (WGS) data from European-ancestry individuals. GTEx gene expression for breast tissue (151 female subjects) and prostate tissue (221 male subjects) were normalized using the Probabilistic Estimation of Expression Residuals (PEER) [42], which adjusts for potential confounding variables. We used 30 PEER factors for our downstream model building based on the GTEx recommendation for the breast and prostate tissue. We collected the whole-genome sequencing data for these subjects and filtered out genetic variants with MAF < 0.01, missingness > 10% and HWE ($P < 1 \times 10^{-6}$). We retained variants located within ±1 Mb of gene transcription start sites (TSS) and overlapping tCRA-prioritized SNVs for TWAS predictive model building. TWAS analysis is a 2-step protocol. In the first step, we trained a predictive model using GTEx expression and genotypes. Particularly, the cis-variants (1Mb flanking region) are used as input terms and the elastic-net is used as the predictive model (Equation 3):

$$E_g \sim \sum_i^p w_{i,g} X_i + \varepsilon$$

(3)

where $E_g$ a vector representing the gene expression of the $g$-th gene; $X_i$ is a vector illustrating the genotype of the $i$-th variant ($X_i$=0, 1, or 2) and $p$ represent the total number of variants in the 1M flanking region for the $g$-th gene; $w_{i,g}$ is the effect size of $i$-th variant for $g$-th gene, and $\varepsilon$ is the residual with $\sigma_e^2$ as its variance. The elastic-net penalty, a mixture of $L_1$ and $L_2$ regularizers is used optimize ($w_{i,g}$) for SNVs.

In the second step, the above trained weights $\hat{w}_{i,g}$ will be applied to the summary statistics of GWAS data (Equation 4):

$$Z_g \approx \sum_{i \in \text{Model}_g} \hat{w}_{i,g} \frac{\hat{\sigma}_i}{\hat{\sigma}_g} \frac{\hat{\beta}_i}{se\left(\hat{\beta}_i\right)}$$

(4)

where, the $Z$ score, $Z_g$, estimates the association between predicted gene expression and cancer risk; $\hat{w}_{i,g}$ is the weight of genetic variant $i$-th for predicting the expression of gene $g$. $\hat{\beta}_i$ and $\text{se}(\hat{\beta}_i)$ are the GWAS-reported regression coefficients and its standard error for $i$-th variant; and $\hat{\sigma}_i$ and $\hat{\sigma}_g$ are the estimated variances of $i$-th variant and the predicted expression of gene $g$, respectively. Assuming $Z_g$ follows a normal distribution, we estimated $P$ values for all genes, and the ones that are lower than 0.05 after Bonferroni-correction are reported as significant.

## Functional evaluation of TWAS genes

To validate the potential functionality of TWAS significant genes, we assessed these genes through three aspects. First, we evaluated their biological relevance by examining gene essentiality using data from the Cancer Dependency Map (DepMap) [43] which provides CERES scores derived from CRISPR-Cas9 knockout screens (DepMap 23Q4). These scores measure the impact of gene knockdown on cancer cell proliferation, with a common cutoff of a median CERES score below −0.5 indicating essential roles in cell proliferation. Next, we assessed the therapeutic relevance of TWAS-identified genes by mapping them to known drug–target pairs using three major databases: DrugBank [44], ChEMBL [45], and the Therapeutic Target Database (TTD) [46]. DrugBank contains over 500,000 drugs; ChEMBL provides data from more than 2 million chemical molecular bioactivity assays; and TTD catalogs over 400 approved drug targets and more than 1,000 targets currently in clinical trials. We integrated drug–gene interaction data from these resources and identified TWAS-significant genes with existing drug associations, highlighting their potential for therapeutic recommendations or drug repurposing. Finally, we evaluated the disease relevance of these genes using DisGeNET [47], a curated and literature-mined database of disease–gene associations. Genes overlapping with entries in DisGeNET were considered to have strong evidence of association with diseases, supporting their potential biological and clinical significance.

## Supporting information

**S1 Fig. TL-breast and TL-prostate model architecture, adapted from Enformer.** The TL model is segmented into five distinct blocks, each composed of multiple layers. The output shapes, excluding the batch dimensions, are denoted by tuples situated to the right of each block. 'C' denotes the number of channels, amounting to 1,536. The TL models is characterized by a single output head, fine-tuned to cater to its designated TF tracks.
(TIF)

**S2 Fig. Track-track Pearson correlations.** A) Track-track pairwise Pearson correlations across 1,937 held-out test sequences for TFs in breast tissue; B) Same as (A), stratified by cell line; C) Track-track pairwise Pearson correlations across 1,937 held-out test sequences for TFs in prostate tissue; D) Same as (C), stratified by cell line.
(TIF)

**S3 Fig. Box plots of ChIP-seq quality metrics stratified by Pearson correlation performance (low, moderate, and high) across 257 breast tissue tracks.** A) FastQC score, reflecting overall sequencing quality. B) Uniquely mapped read ratio, representing the proportion of reads that align uniquely to the reference genome. C) PCR bottleneck coefficient (PBC), measuring library complexity and PCR duplication bias. D) Peaks Fold Change over 10, indicating the proportion of peaks with at least ten-fold enrichment over background signal, reflecting peak strength. E) Fraction of reads in peaks (FRiP), quantifying signal enrichment within called peak regions. F) Union DNase I hypersensitive site (DHS) ratio, representing the fraction of reads overlapping open chromatin regions.
(TIF)

**S4 Fig. Box plots of ChIP-seq quality metrics stratified by Pearson correlation performance (low, moderate, and high) across 375 prostate tissue tracks.** A) FastQC score. B) Uniquely mapped read ratio. C) PCR bottleneck coefficient (PBC). D) Peaks Fold Change over 10. E) Fraction of reads in peaks (FRiP). F) Union DNase I hypersensitive site (DHS) ratio.
(TIF)

**S5 Fig. TL-breast and TL-prostate model performance on validation and test dataset.**
(TIF)

**S6 Fig. Loss and performance of the baseline CNN model trained on breast TF ChIP-seq data.**
(TIF)

**S1 Table. TF ChIP-seq tracks (n = 275) from breast cell lines obtained from Cistrome.**
(XLSX)

**S2 Table. TF ChIP-seq tracks (n = 357) from prostate cell lines obtained from Cistrome.**
(XLSX)

**S3 Table. Track–track Pearson correlations for breast TFs across 1,937 test sequences.**
(XLSX)

**S4 Table. Track–track Pearson correlations for prostate TFs across 1,937 test sequences.**
(XLSX)

**S5 Table. TF ChIP-seq tracks from the original Enformer model.**
(XLSX)

**S6 Table. Correlation between Enformer and TL-prostate predictions for each transcription factor.**
(XLSX)

**S7 Table. Functional enrichment results of SNV subsets (1M, 1.5M, 2M) prioritized by TL and Enformer models.**
(XLSX)

**S8 Table. Significant TWAS genes from base models in breast and prostate.**
(XLSX)

**S9 Table. List of TWAS-significant genes identified by TL and Enformer models across three SNV subsets in breast and prostate cancer.**
(XLSX)

**S10 Table. Summary of TWAS gene discovery for 1M, 1.5M, and 2M TL- and Enformer-prioritized SNV subsets in breast and prostate cancer.**
(XLSX)

**S11 Table. DepMap CERES dependency scores (version 23Q4) for significant genes identified from TL-breast models.**
(XLSX)

**S12 Table. DepMap CERES dependency scores (version 23Q4) for significant genes identified from TL-prostate models.**
(XLSX)

**S13 Table. Druggable genes identified from TL-prioritized 1M SNVs and associated approved or investigational drugs derived from drug databases.**
(XLSX)

**S14 Table. List of ten representative functionally annotated cell lines.**
(XLSX)

**S15 Table. List of risk TFs for breast and prostate cancer.**
(XLSX)

## Acknowledgments

The computational infrastructure was partly supported by a Canada Foundation for Innovation JELF grant (36605) to Q.L. (Q. Long).

## Author contributions

**Conceptualization:** Qing Li, Quan Long.

**Data curation:** Qing Li, Dinghao Wang, Zilong Zhang, Deshan Perera, Zhishan Chen, Wanqing Wen, Xingyi Guo, Quan Long.

**Formal analysis:** Qing Li, Dinghao Wang, Deshan Perera, Zhishan Chen, Wanqing Wen, Xingyi Guo, Quan Long.

**Methodology:** Qing Li, Dinghao Wang, Zilong Zhang, Deshan Perera, Zhishan Chen, Wanqing Wen, Xingyi Guo, Quan Long.

**Resources:** Jun Yan, Xiao-Ou Shu, Wei Zheng, Xingyi Guo, Quan Long.

**Software:** Qing Li.

**Supervision:** Xingyi Guo, Quan Long.

**Visualization:** Qing Li.

**Writing – original draft:** Qing Li, Dinghao Wang, Zilong Zhang, Deshan Perera, Zhishan Chen, Wanqing Wen, M. Ethan MacDonald, Weijia Cai, Jun Yan, Xiao-Ou Shu, Wei Zheng, Xingyi Guo, Quan Long.

**Writing – review & editing:** Qing Li, Dinghao Wang, Zilong Zhang, M. Ethan MacDonald, Weijia Cai, Quan Long.

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
