## [Decision Letter · Decision Letter 0]

11 Dec 2025

PGENETICS-D-25-01035

Tissue-specific transfer learning improves functional variant and therapeutic target discoveries in breast and prostate cancer

PLOS Genetics

Dear Dr. Li,

Thank you for submitting your manuscript to PLOS Genetics. After careful consideration, we feel that it has merit but does not fully meet PLOS Genetics's publication criteria as it currently stands. Therefore, we invite you to submit a revised version of the manuscript that addresses the points raised during the review process.

We look forward to receiving your revised manuscript.

Kind regards,

Xiang Zhou, Ph.D.

Academic Editor

PLOS Genetics

Xiaofeng Zhu

Section Editor

PLOS Genetics

Aimée Dudley

Editor-in-Chief

PLOS Genetics

Anne Goriely

Editor-in-Chief

PLOS Genetics

**Additional Editor Comments:**

Please note that Reviewer #1's detailed comments are included in the attached file.

**Journal Requirements:**

At this stage, the following Authors/Authors require contributions: Qing Li, Deshan Perera, Zhishan Chen, Wanqing Wen, Zilong Zhang, Dinghao Wang, Jun Yan, Xiao-Ou Shu, Wei Zheng, Xingyi Guo, and Quan Long. Please ensure that the full contributions of each author are acknowledged in the "Add/Edit/Remove Authors" section of our submission form.

The list of CRediT author contributions may be found here: https://journals.plos.org/plosgenetics/s/authorship#loc-author-contributions

4) We notice that your supplementary Figures are included in the manuscript file. Please remove them and upload them with the file type 'Supporting Information'. Please ensure that each Supporting Information file has a legend listed in the manuscript after the references list.

2) If any authors received a salary from any of your funders, please state which authors and which funders..

**Reviewers' comments:**

Reviewer's Responses to Questions

**Comments to the Authors:**

Reviewer #1: Please see the attachment word file.

Reviewer #2: Li, et al. adapt the Enformer model for variant effect prediction using transfer learning in order to focus on transcription factor tracks from breast and prostate cancer cell lines. They then evaluate the predicted variant effects from this new model compared to the original Enformer model.

Major comments

1) Clarify what training data was used.

The authors state:

“For breast cancer, we obtained tracks for 18 TFs … yielding a total of 275 tracks”

“For prostate cancer, we selected 17 TFs … totaling 357 tracks”

The Supplementary Tables indicate multiple cell lines were used for each TF, but it looks like many of the tracks a biological or technical replicates. How many biologically distinct tracks were used and how were replicates handled?

In Figure 2, the performance often varies widely across many tracks for the same TF. Is this biological or technical?

2) Demonstrate contribution of transfer learning

The authors seek to demonstrate the value of adapting a pretrained Enformer model using transfer learning. While they demonstrate that training on their specific cancer tracks is useful, they don’t directly demonstrate the benefit of transfer learning compared to training a new model without the pretraining step. Does using transfer learning instead of a new set of parameters just reduce training time or meaningfully increase predictive performance?

Reviewer #3: This paper adapted and fine-tuned existing deep learning tool Enformer to breast and prostate cancer using 275 and 357 tissue-specific transcription factor (TF) ChIP–seq tracks. Further application of the fine-tuned Enformer include tissue-specific prioritization of variants using these variants for improving TWAS. It is claimed the proposed approach enhances regulatory variant prioritization and improve susceptibility gene discovery in a tissue-specific manner. I have some comments as follows,

1. As so many TFs are involved in calculating the tCRAs for variant prioritization. However, the formula on line 400 only indicate one variant and one TF. Please clarify how the variant score is calculated considering so many TFs are collected and for each TF, there are multiple biological replicates (tracks). Please formularize these in a more rigorous way.

2. More formula for loss function for TF-breast and TF-prostate are needed. Only mention “The model was trained to predict ChIP-seq signals using a Poisson negative log-likelihood loss function is not enough.” A follow-up question is, as processed ChIP-seq signals are not count data, why use Poisson distribution?

3. One key concern is the prediction performance is very low (Fig2) but the downstream analysis of variant effect is based on the difference of accurate prediction between ref and alt, which make the variant effect analysis less reliable.

**Have all data underlying the figures and results presented in the manuscript been provided?**

Reviewer #1: Yes

Reviewer #2: Yes

Reviewer #3: Yes

PLOS authors have the option to publish the peer review history of their article (what does this mean?). If published, this will include your full peer review and any attached files.

Reviewer #1: **Yes:** Yichao Zhou

Reviewer #2: No

Reviewer #3: No

**Figure resubmission:**
---

## [Decision Letter · Decision Letter 1]

16 Feb 2026

PGENETICS-D-25-01035R1

Tissue-specific transfer learning improves functional variant and therapeutic target discoveries in breast and prostate cancer

PLOS Genetics

Dear Dr. Li,

Thank you for submitting your manuscript to PLOS Genetics. After careful consideration, we feel that it has merit but does not fully meet PLOS Genetics's publication criteria as it currently stands. Therefore, we invite you to submit a revised version of the manuscript that addresses the points raised during the review process.

We look forward to receiving your revised manuscript.

Kind regards,

Xiang Zhou, Ph.D.

Academic Editor

PLOS Genetics

Xiaofeng Zhu

Section Editor

PLOS Genetics

Aimée Dudley

Editor-in-Chief

PLOS Genetics

Anne Goriely

Editor-in-Chief

PLOS Genetics

**Journal Requirements:**

At this stage, the following Authors/Authors require contributions: Qing Li, Dinghao Wang, Zilong Zhang, Deshan Perera, Zhishan Chen, Wanqing Wen, M. Ethan MacDonald, Weijia Cai, Jun Yan, Xiao-Ou Shu, Wei Zheng, Xingyi Guo, and Quan Long. Please ensure that the full contributions of each author are acknowledged in the "Add/Edit/Remove Authors" section of our submission form.

The list of CRediT author contributions may be found here: https://journals.plos.org/plosgenetics/s/authorship#loc-author-contributions

2) We notice that your supplementary Figures are included in the manuscript file. Please remove them and upload them with the file type 'Supporting Information'. Please ensure that each Supporting Information file has a legend listed in the manuscript after the references list.

3) In the online submission form, you indicated that your data will be submitted to a repository upon acceptance. We strongly recommend all authors deposit their data before acceptance, as the process can be lengthy and hold up publication timelines. Please note that, though access restrictions are acceptable now, your entire minimal dataset will need to be made freely accessible if your manuscript is accepted for publication. This policy applies to all data except where public deposition would breach compliance with the protocol approved by your research ethics board. If you are unable to adhere to our open data policy, please kindly revise your statement to explain your reasoning and we will seek the editor's input on an exemption.

4) Please amend your detailed Financial Disclosure statement. This is published with the article. It must therefore be completed in full sentences and contain the exact wording you wish to be published.

**Reviewers' comments:**

Reviewer's Responses to Questions

**Comments to the Authors:**

Reviewer #1: The authors have addressed all my comments and suggestions. I thank them for their valuable contribution to the statistical genetics and broader computational biology communities.

Reviewer #2: I am satisfied with the revised version

Reviewer #3: For comment3, the response from the reviewers "as average correlations around ~0.4

are well within the expected range for TF ChIP-seq prediction and align with previously

published deep-learning models." However, why using TF ChIP-seq for prediction is the first question of concern. If prediction only ~0.4, how to expect predicted TF under ref and alt alleles are confident, and how the difference is confident to evaluate variant effect? As ChIP-seq is usually measured by binding sites (binary), not sure if what is predicted ChIP-seq track in a continuous manner here? Moreover, other deep learning models report on binary peak prediction for evaluating variant effect. More explanation and investigation towards this direction would be helpful.

**Have all data underlying the figures and results presented in the manuscript been provided?**

Reviewer #1: Yes

Reviewer #2: None

Reviewer #3: None

PLOS authors have the option to publish the peer review history of their article (what does this mean?). If published, this will include your full peer review and any attached files.

Reviewer #1: No

Reviewer #2: No

Reviewer #3: No

**Figure resubmission:**
---

## [Decision Letter · Decision Letter 2]

25 Apr 2026

Dear Dr Li,

We are pleased to inform you that your manuscript entitled "Tissue-specific transfer learning improves functional variant and therapeutic target discoveries in breast and prostate cancer" has been editorially accepted for publication in PLOS Genetics. Congratulations!

Yours sincerely,

Xiang Zhou, Ph.D.

Academic Editor

PLOS Genetics

Xiaofeng Zhu

Section Editor

PLOS Genetics

Aimée Dudley

Editor-in-Chief

PLOS Genetics

Anne Goriely

Editor-in-Chief

PLOS Genetics

BlueSky: @plos.bsky.social

Comments from the reviewers (if applicable):

Reviewer's Responses to Questions

**Comments to the Authors:**

Reviewer #3: None

**Have all data underlying the figures and results presented in the manuscript been provided?**

Reviewer #3: None

PLOS authors have the option to publish the peer review history of their article (what does this mean?). If published, this will include your full peer review and any attached files.

Reviewer #3: No

**Data Deposition**

http://datadryad.org/submit?journalID=pgenetics&manu=PGENETICS-D-25-01035R2

**Press Queries**

---

## [Editor Report · Acceptance letter]

PGENETICS-D-25-01035R2

Tissue-specific transfer learning improves functional variant and therapeutic target discoveries in breast and prostate cancer

Dear Dr Li,

We are pleased to inform you that your manuscript entitled "Tissue-specific transfer learning improves functional variant and therapeutic target discoveries in breast and prostate cancer" has been formally accepted for publication in PLOS Genetics! Your manuscript is now with our production department and you will be notified of the publication date in due course.

With kind regards,

Anita Estes

PLOS Genetics

On behalf of:
